# Chikungunya Replication and Infection Is Dependent upon and Alters Cellular Hexosylceramide Levels in Vero Cells

**DOI:** 10.3390/v17040509

**Published:** 2025-03-31

**Authors:** Joseph Thomas Noble, Kingsley Bimpeh, Michael Anthony Pisciotta, Judith Mary Reyes Ballista, Kelly Marie Hines, Melinda Ann Brindley

**Affiliations:** 1Department of Infectious Diseases, College of Veterinary Medicine, University of Georgia, Athens, GA 30602, USA; jtn84556@uga.edu (J.T.N.); michael.pisciotta@uga.edu (M.A.P.); reyes.judith94@gmail.com (J.M.R.B.); 2Department of Chemistry, University of Georgia, Athens, GA 30602, USA; bimpeh.kingsley@bcg.com (K.B.); kelly.hines@uga.edu (K.M.H.); 3Department of Population Health, College of Veterinary Medicine, University of Georgia, Athens, GA 30602, USA

**Keywords:** chikungunya virus, alphavirus, lipidomics, hexosylceramide, fatty acid synthesis

## Abstract

Chikungunya virus (CHIKV), a mosquito-borne alphavirus, causes significant global morbidity, including fever, rash, and persistent arthralgia. Utilizing untargeted lipidomics, we investigated how CHIKV infection alters host cell lipid metabolism in Vero cells. CHIKV infection induced marked catabolism of hexosylceramides, reducing their levels while increasing ceramide byproducts. Functional studies revealed a reliance on fatty acid synthesis, β-oxidation, and glycosphingolipid biosynthesis. Notably, inhibition of uridine diphosphate glycosyltransferase 8 (UGT8), essential for galactosylceramide production, significantly impaired CHIKV replication and entry in Vero cells. Sensitivity of CHIKV to UGT8 inhibition was reproduced in a disease-relevant cell line, mouse hepatocytes (Hepa1-6). CHIKV was also sensitive to evacetrapib, a cholesterol ester transfer protein (CETP) inhibitor, though the mechanism of inhibition appeared independent of CETP itself, suggesting an off-target effect. These findings highlight specific lipid pathways, particularly glycosphingolipid metabolism, as critical for CHIKV replication and further refine our understanding of how CHIKV exploits host lipid networks. This study provides new insights into CHIKV biology and suggests that targeted investigation of host lipid pathways may inform future therapeutic strategies.

## 1. Introduction

Chikungunya virus (CHIKV, family *Togaviridae*, genus *Alphavirus*) is a mosquito-borne pathogen transmitted by *Aedes aegypti* and *Aedes albopictus*, the same vectors that transmit dengue, Zika, and yellow fever viruses [1]. Infection can cause fever, rash, and arthralgia, with incapacitating joint pain that can last for months [2,3]. Immunocompromised patients present with more severe disease symptoms, including myocarditis and encephalitis [4,5]. Periodic CHIKV outbreaks occurred in the Americas in the past, but the most recent epidemic was caused by the introduction of CHIKV into the Caribbean in 2013. This resulted in a large-scale outbreak with more than 1 million suspected cases within the year [6,7]. The long-term sequelae associated with CHIKV infection result in a significant economic burden [8], and it remains endemic throughout the Americas. Global climate change is expected to increase the geographic range of the mosquito vectors and, therefore, the number of people potentially exposed to arbovirus infection [9,10]. A new vaccine was recently approved [11], and hopefully, it, along with vector control strategies, could lead to a decrease in future outbreaks.

Viruses require a host cell to provide the necessary building blocks to reproduce, including nucleic acids, amino acids, and lipids. While most research focuses on the role of genome replication and protein production, lipids are essential for all steps in the CHIKV reproduction cycle. CHIKV virions are enveloped viruses; thus, to deliver their genome into the cytoplasm and initiate infection, the viral lipid envelope must fuse with a cellular membrane [12,13,14]. Efficient alphavirus fusion requires cholesterol and sphingolipids in the target cell lipid bilayer [15,16,17]. Genome replication of all positive-sense RNA viruses is associated with lipid membranes [18,19,20]. Alphavirus infection leads to spherule formation, a replication organelle consisting of a single-membrane bulb-shaped bud at the plasma membrane of mammalian cells [21,22,23,24,25,26,27,28]. Studies have yet to identify the cellular proteins and lipids required for spherule formation [27]. The last step of the replication cycle is viral budding, where the alphavirus capsid is enrobed in lipids, stealing them from a cellular membrane.

Cellular lipid levels are highly regulated to maintain equilibrium within cells. Viral infection induces many changes within the infected cell, including changes in the lipid profile [29,30,31,32,33]. As positive-sense RNA viruses reorganize membranes while forming replication organelles, additional lipids are produced [34,35]. Lipidomic studies have documented alterations after infection with rhinovirus [36], dengue [37], hepatitis C [38,39], and human coronavirus 229E (HCoV-229E) [40]. Other viruses that do not remodel internal cellular membranes, like influenza [41,42], VSV [43], and HIV [44], also alter specific cellular lipids during infection. To determine how the chikungunya virus alters the cellular lipidome, we performed untargeted lipidomics on whole CHIKV-infected cells to capture the viral-induced changes. CHIKV infection resulted in catabolism of carbohydrate-modified complex lipids called hexosylceramides, dramatically reducing the hexosylceramide levels and increasing the corresponding byproduct, ceramides. Using various small molecule inhibitors known to alter lipid synthesis pathways, we highlight the roles of specific lipids in the different stages of CHIKV replication and posit that this increased hexosylceramide flux may be an antiviral response of the cell.

## 2. Materials and Methods

### 2.1. Materials

HPLC-grade solvents (water, acetonitrile, methanol, and chloroform) and ammonium acetate were purchased from Thermo Fisher Scientific (Waltham, MA, USA). A mixture of sphingolipids, glycerolipids, and phospholipids representative of mammalian lipid composition was prepared as described previously [45] and used as a HILIC retention time reference. Drugs that alter lipid metabolism were purchased from Cayman Chemical (Ann Arbor, MI, USA), MedChem Express (Monmouth Junction, NJ, USA), or Sigma (St. Louis, MO, USA). All drugs were resuspended in DMSO and stored in small aliquots at −20 °C for less than 6 months.

### 2.2. Cell Culture and Viruses

Vero (African green monkey kidney) cells (ATCC CCL-81, Manassas, VA, USA) and Vero cells stably expressing SLAM (VeroS) [46] were maintained at 37 °C and 5% CO_2_ in Dulbecco’s modified Eagle’s medium (DMEM) supplemented with 5% fetal bovine serum. Hepa1-6 (ATCC CRL-1830, Manassas, VA, USA, a kind gift from Dr. Samarchith Kurup, University of Georgia, Athens, GA, USA) were maintained in DMEM supplemented with 10% FBS. Recombinant CHIKV strain 181/25c encoding GFP and E2-tagged nano-luciferase and VSV encoding GFP have been described previously [47,48]. Titers were determined via endpoint titration TCID50 on VeroS cells (Spearman–Karber method).

### 2.3. Viral Infection for Lipidomic Analysis

VeroS cells were grown to 90% confluence in 10 cm plates and mock-infected or infected with CHIKV-GFP or VSV-GFP at an MOI of 0.1 for 1 h. After infection, the media was replaced with DMEM supplemented with 5% FBS, and cells were maintained for 18-24 h until 95% of the cells were GFP positive. Supernatants were removed, and cells were rinsed in PBS, pelleted (800× *g*, 3 min) in PBS, and stored at −80 °C until lipid extraction. Five independent plates of cells were prepared for each condition.

### 2.4. Lipid Extraction

VeroS cells (uninfected, VSV-infected, and CHIKV-infected) were suspended in 0.5 mL of PBS, transferred to a glass tube, and sonicated in an ice bath for 30 min. Lipid extraction was performed using a modified version of the Bligh and Dyer extraction method [49,50,51]. Two milliliters of chilled premade 2:1 chloroform/methanol (*v*/*v*) were added to the samples and vortexed intermittently for 5 min on ice. An additional amount of 0.5 mL each of chloroform and water was added to the suspension in a glass tube and vortexed briefly for a minute. The resulting mixture was centrifuged for 10 min at 2000× *g* and 4 °C. The organic layer was then transferred to a new glass tube and dried in a speed-vac concentrator. The dried lipid extracts were subsequently reconstituted with 1:1 chloroform/methanol (*v*/*v*) for storage at −80 °C.

### 2.5. HILIC-IM-MS

Lipid extracts were analyzed using an online Hydrophilic Interaction Liquid Chromatography (HILIC) coupled with a Waters Synapt XS Travelling Wave Ion-Mobility Mass Spectrometry (TWIMS-MS). Chromatographic separation was performed at 40 °C with Waters Cortecs Ultra Performance Liquid Chromatography (UPLC) HILIC column (2.1 × 100 mm, 1.6 µm) connected to Waters Acquity FTN I-Class Plus UPLC. Mobile phase A (MPA) consisted of 50% acetonitrile and 50% water with 10 mM ammonium acetate, and mobile phase B (MPB) consisted of 95% acetonitrile and 5% water with 10 mM ammonium acetate. Using a fixed flow rate of 0.5 mL/min, lipids were separated using the following gradient method: 1–2 min, 100% MPB; 3–4 min, 60% MPB; 5–7 min, 100% MPB. Lipid extracts were dried in a Savant vacuum concentrator (Thermo Fisher Scientific, Waltham, MA, USA) and prepped at a 15× dilution with MPB. Quality control (QC) samples were prepared by pooling equal volumes from each sample. All prepped samples were stored in an autosampler maintained at 6 °C to maintain sample integrity over the LC-MS run. All experiments were performed with an injection volume of 5 µL.

Mass spectra were acquired in positive and negative ion mode with the following electrospray ionization (ESI) source conditions: capillary voltage, (+/−) 2.0 kV; sampling cone voltage, 40 V; source offset, 4 V, source temperature, 135 °C; desolvation temperature, 500 °C; desolvation gas flow rate, 1000 L/h; cone gas flow rate, 50 L/h. TWIM separations were performed in nitrogen with a gas flow of 90 mL/min, wave velocity of 550 m/s, and wave height of 40 V. TWIM separations were performed in nitrogen with a gas flow of 90 mL/min, wave velocity of 550 m/s, and wave height of 40 V. Mass calibrations were performed with sodium formate over the range of 50–1200 *m*/*z*. Continuum data were collected with a 0.5 s scan time over the range of 50–1200 *m*/*z* with the time-of-flight (TOF) mass analyzer operating in resolution mode (resolution of ~30,000). Leucine-enkephalin was continuously infused over the entire acquisition time for lock mass correction. Ion mobility-organized data-independent acquisition (MS^E^) was performed in the post-ion mobility transfer region of the instrument with a 35–50 eV collision energy ramp.

### 2.6. Data Analysis and Processing

Data processing was performed in Progenesis QI software (v3.0, Waters/Nonlinear Dynamics). Samples were grouped into their respective categories after peak picking and lock-mass correction with the leucine enkephalin lock-mass signal. Principal Component Analysis (PCA) was performed with EZinfo (v3.0, Umetrics) on the data set filtered by an ANOVA *p*-value indicated in the PCA plots. PCA loadings were investigated on the filtered data to identify lipids that showed statistically significant differences among the investigated groups. Lipid identifications were carried out using retention times and accurate mass (<10 ppm) compared with an in-house database developed from LipidPioneer and the Human Metabolome Database (HMDB) [52].

### 2.7. Data Availability

Raw data files and processed result files are available on the MassIVE repository (accession number pending).

### 2.8. Drug Inhibition Assays

Vero cells were seeded in 96-well plates (5 × 10^4^ cells/well) in DMEM+5%FBS. Drugs were diluted in DMEM and added to the wells at the indicated concentrations the following day. CHIKV-E2NLuc-GFP was diluted in DMEM and added to the wells right after drug addition (MOI 0.001). Twenty-four hours following infection, the cells were lysed in Promega NanoGlo assay working reagent, and relative luminescence was measured in a Promega GloMax Explorer. To determine if the drugs inhibited the virus, the luciferase values in the presence of the drugs were compared to the DMSO control. Cell viability was examined in a duplicate plate that lacked viral infection. These cells were lysed with Promega Cell-TiterGlo buffer, and luminesce corresponding to the amount of ATP in the well was measured and compared to the DMSO control.

### 2.9. Virus Entry Assays

VeroS cells were plated in 48-well plates (2.5 × 10^5^ cells/mL). For short-term pre-treatment, 200 µL of DMEM 5% FBS with compounds at 1.5× final concentration was added to the cells for 1 h. Then, 100 µL of CHIKV-GFP stock diluted in DMEM was added, diluting the treatment media to the indicated final concentration. For long-term pre-treatment, confluent VeroS cells were incubated in DMEM 5% FBS with drugs at the indicated concentrations for 18 h, then 100 µL of CHIKV-GFP stock diluted in DMEM was applied. A DMSO-treated, mock-infection well was included to establish a GFP+ gate for flow cytometry. For both short and long-term pre-treatment, infection and entry were allowed to progress for two hours before the media was removed, cells were washed with PBS, and then replaced with 300 µL of DMEM 5% FBS with 10 mM NH_4_Cl to prevent further entry. Infected cells were harvested at 16 hpi to ensure collection before evident CPE; cells were trypsinized, resuspended in DMEM 5% FBS, and fixed at 1:1 *v*/*v* with 3.7% formalin prior to flow cytometry. Cells were analyzed on a Novocyte Quanteon using the following schema: live (SSC-H/FSC-H), single (SSC-A, SSC-H), GFP+ (Count/B530). Ten thousand live, single cells were collected, and the %GFP+ of that population is reported.

### 2.10. Effect of Drugs on CHIKV Multi-Step Infection Titers

Confluent VeroS cells in 48-well plates were treated with 125 µL of treatment with vehicle (DMSO) or the indicated drug at 2× concentration. Immediately after, 125 µL of CHIKV-GFP at the indicated MOI was applied on top of the treatment. Supernatants were collected at 24 hpi and titrated, and cells were harvested via trypsinization and analyzed via flow cytometry to determine the number of virally infected cells. Pre-treated multi-step infection occurred similarly; however, in this case, cells were treated for 18 hpi with DMEM-5% FBS and drugs at the indicated concentrations. This media was then removed, and the regular protocol was followed.

### 2.11. Virucidal Assay

To determine if the indicated compounds were virucidal, CHIKV-GFP stock was diluted in ice-cold DMEM 5% FBS to yield a titer of ~1 × 10^6^ TCID50 units/mL in a total volume of 1 mL. These diluted viral stocks were vortexed and equilibrated on ice. Then, 1 µL of 1000× stock of indicated drugs or DMSO was diluted within the viral samples, vortexed briefly, and immediately placed in a 37 °C water bath and incubated for 1 h. Following incubation, the drug-incubated viral samples were immediately titrated via the previously mentioned endpoint titration (TCID50).

### 2.12. Replicon Generation and Assay

To generate the CHIKV replicon, a CMV promoter, *eGFP*, and *NLuc-hPEST* were cloned into an infectious clone. First, the structural protein cassette was replaced with nano-luciferase-hPEST in the plasmid pSinRep5-181/25ic-GFP, producing GFP from an additional sub-genomic RNA engineered between the non-structural and structural protein cassettes, described in [48]. To obviate the need for in vitro transcription, a CMV promoter was cloned upstream of the SP6 promoter. The resulting plasmid will be referred to as pCHIKVreplicon-eGFP-NLucP, deposited in Addgene. All cloning was performed using In-Fusion ligation per the manufacturer’s protocol (Takara Biosciences, San Jose, CA, USA).

To measure the effects of the drug on the replicon, semi-confluent (~70%) VeroS cells were transfected with pCHIKVreplicon-eGFP-NLucP using JetOptimus transfection reagent per manufacturer’s protocols (Polyplus, New York, NY, USA). Four hours following transfection, cells were washed, and DMEM-5% FBS containing the indicated drugs was added. Luciferase levels were determined 24 h following drug addition using NanoGlo Luciferase working reagent per the manufacturer’s protocols (Promega, Madison, WI, USA).

### 2.13. Statistics

Data analysis and graphs were produced using GraphPad Prism (v10.4.0, Domatics, Boston, MA, USA). An unpaired parametric Student’s *t*-test was performed to determine the significance between the two groups. For normalized data, a Welch’s correction was used. For logarithmic data, values were first natural log (ln) transformed and then analyzed with *t*-tests. An ordinary one-way ANOVA with multiple comparisons was used to evaluate statistical differences among more than two groups with non-normalized data.

## 3. Results

### 3.1. CHIKV Infection Alters Cellular Lipid Levels

To identify lipid changes induced by CHIKV infection, Vero cells infected with CHIKV strain 181/25c that encodes GFP were harvested when cells were >95% GFP positive. Following lipid extraction, a mass-spectrometric method with high resolution for lipid species, HILIC-IM-MS, analyzed the lipid contents of each sample (Figure 1A). Principal component analysis (PCA) compared mass spectrometry-derived lipid profiles of mock-infected cells to both vesicular stomatitis virus (VSV) infected cells and CHIKV-infected cells (Figure 1A). Positive and negative ion-mode mass spectrometry data for the three conditions (mock-infected, CHIKV-infected, and VSV-infected) produced three distinct lipid profiles via multivariate PCA analysis (Figure 1B). The lipid classes significantly contributing the most variance between the samples were called and identified. The most significant differences included ceramides (Cer) and phospholipids. CHIKV infection was associated with increased ceramide levels and a corresponding decrease in hexosylceramides (HexCer), indicating hexosylceramide catabolism (Figure 1C–F). This catabolic signature was specific for CHIKV infection, as VSV infection did not significantly increase ceramide levels and only modestly reduced hexosylceramides (Figure 1F).

Focusing on the changes induced during CHIKV infection, we find that many lipids have been significantly altered (Figure 2A). The most abundant phospholipids within the cell, phosphatidylcholine (PC) and phosphatidylethanolamine (PE), while contributing to variance between mock and CHIKV per the PCA analysis, were not significantly altered by infection (Figure 2B,C). The intermediate metabolite diacylglycerol (DG) was also not significantly altered (Figure 2D). Infection was associated with decreases in two phosphatidyglycerol (PG) species, PG(36:1) and PG(36:2) (Figure 2E). PG is an anionic phospholipid and the precursor for many bis-phospholipids, such as the mitochondrial lipid component cardiolipin [53]. Most variation between mock and CHIKV-infected datasets is derived from hexosylceramides and ceramides (Figure 2F,G). Among the ceramide class, two odd-chain ceramides, Cer(d18:1/21:0) and Cer(d18:1/19:0), and the even-chain Cer(d18:1/24:0) had higher abundance within CHIKV cells. Cer(d18:1/16:0) was the only ceramide seen to decrease in infection. Among hexosylceramides, which are the precursors for the myelin and lipid-raft-associated lipids known as sulfatides and gangliosides [54,55,56], there were decreases in the even-chain 24:1, 22:0, 18:0, and 16:0 HexCer(d18:1). Linked increases in ceramides and decreases in hexosylceramides suggest either increased hexosylceramidase activity or decreased hexosyltransferase activity in CHIKV infected cells [57]. In general, these changes suggest mild perturbances in the cellular lipidome due to infection with CHIKV, without some of the broad changes typically observed with positive-sense viruses, such as increases in PC due to endoplasmic-reticulum associate double-membrane vesicle (DMV) formation [31,40,58]. The observed lipid changes may reflect the novel plasma membrane-associated replication spherules observed in alphavirus replication [59].

### 3.2. Alteration of Lipid Regulatory Elements Disturbs CHIKV Replication

The lipidomic data highlighted cellular lipid changes that occur during infection, but they cannot determine if these changes are due to the virus stimulating these changes for its benefit or if the changes are a cellular response trying to limit viral replication. To further explore the lipids required for optimal CHIKV replication, we treated cells with inhibitors that target lipid regulatory elements and monitored CHIKV replication by measuring NLuc activity encoded by the virus (Figure 3A). First, we looked at compounds known to alter cellular lipid pathways broadly. MK-2206 is an allosteric Akt inhibitor [60]. Akt is a protein within the PI3K/Akt/mTOR pathway, a pathway that stimulates many cellular responses, especially lipogenesis [61]. Via Akt-inhibition, MK-2206 treatment indirectly inhibits mTOR-mediated sterol-regulatory binding protein 1 (SREBP1), an inducer of fatty acid synthesis, and activates sterol regulatory SREBP-2, an inducer of cholesterol metabolism [62,63]. Peroxisome proliferator-activated receptors (PPARs) are transcription factors that promote the transcription of genes involved in lipid metabolism and energy production [64]. Rosiglitazone activates PPARγ. Liver X receptors (LXRs) are transcription factors that are master regulators of cholesterol and lipid homeostasis [65]. GW3965 is a synthetic LXR agonist and alters cholesterol levels in cells [66]. GW3965 and rosiglitazone reduced CHIKV replication in a dose-dependent manner; however, MK-2206 did not alter luciferase levels (Figure 3C). Each of these inhibitors affects transcription factors that have multiple targets, making the specific effect difficult to identify, though previous work suggests they both alter cholesterol levels within the cells.

### 3.3. CHIKV Replication Requires Active Fatty Acid Synthesis and β-Oxidation

Fatty acid synthase (FASN) converts malonyl-CoA into palmitate (Figure 4A), which can then be added to proteins through palmitoylation. Palmitoylation of CHIKV non-structural protein 1 (nsP1) is needed to correctly localize nsP1 to cholesterol-rich microdomains in the plasma membrane for optimal replication efficiency [21,67]. Previous studies demonstrated that FASN activity was required for CHIKV replication [68]. It was, therefore, not surprising to find that FASN inhibitors reduce CHIKV replication (Figure 4B). Unsurprisingly, inhibitors of enzymes upstream of fatty acid synthesis, for example, SLC25, ACLY, and ACC, were, for the most part, highly effective at inhibiting CHIKV replication. SLC25A1 serves as the starting point for fatty acid biosynthesis, enabling the export of citrate from the mitochondria to the cytosol [69]. Immediately downstream, citrate is cleaved by ATP-citrate lyase (ACLY), yielding acetyl-CoA; the differential response of the ACLY inhibitors SB204990 and ETC1002 is unclear, as the mechanisms of action for the drugs have not been elucidated, but it should be noted that ACLY inhibition will also deplete acetyl-CoA necessary for cholesterol production [70,71]. Acetyl-CoA carboxylase (ACC-1), carboxylates acetyl-CoA to malonyl-CoA; each direct inhibitor of ACC-1 shows some degree of dose-responsive inhibition of CHIKV. ACC-1 can be inhibited by AMPK [72], thus, mild inhibition of CHIKV by the AMPK activator PF-06409577 was expected. In the next step, the large, multimeric enzyme fatty acid synthase (FASN) coordinates the addition of acetyl-CoA to malonyl-CoA, which, over seven catalytic steps, yields the fatty acid palmitate (C16:0). CHIKV was sensitive to GSK2194069 and TVB-2640, both of which inhibit the keto-acyl reductase domain of FASN [73,74]. CHIKV was also found to be sensitive to inhibition of the long-chain fatty acyl-CoA synthase (ACSL), which activates fatty acids produced from FASN by forming a thioester bond with Coenzyme A. Triascin C, which inhibits ACSL1, 3, and 4, was an effective inhibitor, while a specific ACSL4 inhibitor, PRGL493, had no effect, suggesting dependence on ACSL1 and/or ACSL3. ACSL1 is the dominant isoform in liver and kidney cells, thioesterifying CoA to medium to long-chain fatty acids (C12-C20) [75]. On the other hand, ACSL4, the target of PRGL493, specifically thioesterifies arachidonic acid (C20:4) to arachidonoyl-CoA, which is integral for eicosanoid synthesis and ferroptosis [76,77,78,79,80].

In addition to active fatty acid synthesis, CHIKV is highly reliant on the active breakdown of fatty acids, known as β-oxidation. β-oxidation breaks down fatty acids derived from lipid droplets via lipolysis (Figure 5A). This reliance on beta-oxidation might explain why CHIKV is dependent, in part, on functional lipase activity. Functional diacylglycerol (DAG) lipolysis by the hormone-sensitive lipase (HSL), but not triacylglycerol (TAG) lipolysis, is critical for CHIKV infection, as shown by sensitivity to CAY10499 and tolerance to Atglistatin and NG-497 (Figure 5B). Interestingly, inhibition of diacylglycerol O-acyltransferase 1 (DGAT1) inhibits CHIKV, suggesting that either lipid droplet formation is crucial for CHIKV or that increased DAG levels, as seen with HSL and DGAT inhibition, are antiviral.

### 3.4. CHIKV Sensitivity to Galactosylceramide Modulation

The lipidomic data identified an increased level of ceramides and a large decrease in hexosylceramides during CHIKV infection. Ceramides are complex lipids whose biosynthesis starts in the ER by serine palmitoyltransferase (SPT), converting palmitoyl-CoA to 3-ketosphinganine. Eventually, ceramide synthase (CS) adds fatty acyl-CoA forming ceramide (Figure 6A). Ceramide can be converted into several glycosphingolipids. Cells treated with SPT and CS inhibitors that should block upstream of ceramide production did not affect CHIKV replication, nor did inhibitors that prevent the production of glucosylceramides(GlcCer) (Eliglustat, DNJ, AMP-DNM) (Figure 6B). However, uridine diphosphate glycosyltransferase 8 inhibitor 19 (UGT8-In19), which blocks the production of galactosylceramides (GalCer), decreased CHIKV replication (Figure 6B). D609 inhibits both the sphingomyelin synthase and also blocks phospholipase C [81,82]. D609 treatment decreased CHIKV replication (Figure 6B). Blocking sphingosine to sphingosine-1-phosphate (S-1-P) also modestly reduced CHIKV replication (Figure 6B), similar to findings by other groups [83].

### 3.5. CHIKV Relies on Cholesterol Synthesis, Modification, and Transport

Cholesterol is an important component of the cell lipid membrane and is concentrated in microdomains in the plasma membrane called lipid rafts. While the liver produces most of the cholesterol within a host, all cells can synthesize cholesterol de novo through the mevalonate (MVA) pathway. Cells can also obtain dietary cholesterol through receptor-mediated endocytosis of very low-density lipoproteins, low-density lipoproteins, and high-density lipoproteins (VLDL, LDL, HDL). LDLs are composed of a phospholipid and cholesterol shell surrounding a hydrophobic core of triglycerides and cholesterol esters. Once bound to the LDL receptor, they are trafficked to the late endosome/lysosome compartment where cholesterol esters are hydrolyzed and free cholesterol is released into the cytoplasm (Figure 7B). De novo cholesterol production is a highly regulated, multi-step process [84]. HMG-CoA reductase (HMGCR) converts HMG-CoA to mevalonate, which is the rate-limiting step in cholesterol biosynthesis (Figure 7B). Statins are structural analogs of HMG-CoA and can compete with HMGCR, reducing cholesterol production. Simvastatin reduced CHIKV-NLuc levels by 35% at the highest concentration (Figure 7C). Cetaben, which inhibits cholesterol, triglyceride, and cholesterol ester synthesis [85,86], reduced CHIKV reporter levels by 60% (Figure 7C). Cholesterol uptake within cells requires Niemann-Pick (NPC1) activity in the lysosome to transport free cholesterol to the cytoplasm (Figure 7A). NPC1 inhibitor, U18666A, modestly reduced CHIKV replication (Figure 7C). When cholesterol levels within the cell are high, ACAT will add a fatty acid moiety, producing cholesterol esters stored in cellular lipid droplets. Cl-976, an ACAT inhibitor, reduced CHIKV replication.

Cholesterol ester transfer proteins (CETP) are secreted proteins that facilitate the movement of triglycerides and cholesterol esters between lipoproteins [87]. They exchange triglycerides from VLDL and LDL with cholesterol esters from high-density lipoprotein (HDL) or vice versa (Figure 8A). The CETP inhibitors dalcetrapib and evacetrapib both reduced CHIKV replication while not altering cell viability (Figure 8B).

### 3.6. Mechanisms of Inhibition

Compounds that proved inhibitory in a dose-responsive manner, and which have not been examined in previous studies, were further evaluated to determine whether they decreased infectious virus titers and spread in VeroS cells. Evacetrapib, D609, and UGT8-In19 significantly reduced the percent of infected cells over 24 h (Figure 9A); however, titer decreases were only observed with evacetrapib and UGT-In19 (Figure 9B). Co-incubation of the drugs and virus had no effect on viral titers, suggesting the measured response was not due to virucidal effects (Figure 9C). To determine which step(s) in the viral life cycle were inhibited, we monitored virus entry and genome replication. Cells were either pre-treated for short- (1 h) or long time periods (18 h) and infected with CHIKV-GFP. After a two hour infection period, the virus-drug solution was removed and media containing ammonium chloride was added to the cells to prevent additional entry events. In the short-term entry assay, only D609 yielded a strong 50% reduction, while CAY10499 caused a modest but significant decrease in entry (Figure 9D). In the long-term treatment, UGT8-In19, D609, and CAY10499 significantly decreased CHIKV-GFP entry, suggesting they both require time to alter the cells to block CHIKV entry, whereas D609 acts more rapidly (Figure 9E).

To examine the effects of these drugs on replication, cells were transfected with a CHIKV replicon (pCHIKV-Replicon-eGFP-NLucP) and treated with drugs three hours post transfection. The palmitoylation inhibitor, 2-bromopamitate (2-BP), strongly decreased replicon expression, as previously observed [67]. Evacetrapib and UGT8-In19 significantly decreased CHIKV replicon nano-luciferase levels by roughly 90% (Figure 9F).

To determine whether CHIKV dependence on UGT8 activity is conserved in a disease-relevant cell line, we infected immortalized mouse hepatocytes (Hepa1-6) cells with CHIKV-GFP, followed by treatment with vehicle or UGT8-In19. Multiple models of CHIKV find high levels of replication in the liver, supporting hepatocytes as a disease-related cell type [88,89,90]. UGT8-In19 treatment significantly decreased CHIKV spread at 4in Hepa1-6 cells (Figure 10A). Supernatants collected from the Hepa1-6 cells demonstrate that the UGT8 inhibition leads to more than a ten-fold decrease in titers (Figure 10B).

From our array of inhibitors, evacetrapib was one of the most potent. This is the first known example of a virus being sensitive to a CETP inhibitor and viral sensitivity to an inhibitor of a media-resident protein. We set about determining how an inhibitor of a media-resident protein reduces the replication and production of infectious virus. To determine whether this inhibition was specific to lipoprotein-containing media, we treated infected cells in lipoprotein-depleted media. Evacetrapib treatment in both FBS-supplemented media and lipoprotein-depleted (LD) FBS-supplemented media decreased CHIKV-GFP spread and titers in a multi-step infection (Figure 11), suggesting a mechanism of inhibition independent of CETP.

## 4. Discussion

Positive-sense viruses are known to induce drastic changes in the lipidomes of the cells they infect [32,38,91]. Prior to this study, an understanding of how CHIKV interacts with and alters the cellular lipid environment was constrained to (1) its dependence on fatty acid palmitoylation for replication [21], (2) affinity of the replication complex for negative phospholipids, detergent-resistant microdomain, and increased membrane thickness at the spherule: membrane interface [92,93], and (3) dependence on sterol regulatory binding element 1 (SREBP) regulated lipid biosynthesis genes stearoyl-CoA desaturase (SCD1) and FASN [68]. Our lipidomic analysis suggests no drastic change in the abundance of phospholipids within the cells, in contrast to what has been seen in multiple positive-sense virus-infected cellular lipidomes [12,31]. The most drastic change we observed was a shift from glycosylated ceramide to ceramides, suggesting increased hexosylceramidase activity, a condition typically associated with induction of the unfolded protein response (UPR) and autophagy [94,95]. The term “hexosylceramidase” includes the catabolic enzymes glucosyl- and galactosyl-ceramidase, the major forms of which are lysosome-bound enzymes whose deficiency is well characterized to result in lysosomal storage diseases (LSDs) [96]. The UPR-regulated CHOP protein is a transcription factor for GBA1, the gene encoding glucosylceramidase (GCase), and increased expression of GCase has been observed in some but not all cases of induced UPR [97,98]. Associated conditions that might induce expression of galactosylceramidase (GALC) are less well characterized, though KO of the autophagic protein Atg7 has been shown to lead to increased GALC in a possible compensatory manner [99]. Studies indicate that chikungunya induces autophagy, ER-stress, and has proteins that limit an incumbent UPR response, and thus, this presents a possible mechanism for the observed flux in hexosylceramides as a consequence of CHIKV-associated cellular response consisting of increased lysosomal degradation and protein synthesis restriction [100,101,102,103,104].

Previous insight into the effects of CHIKV infection on cellular glycosylation and ceramides indicated that the ceramide-derived sphingosine-1-phosphate (S-1-P) biosynthetic enzyme, sphingosine-kinase-2 (SK2), is a required host factor in CHIKV infection of multiple cell lines [83,105]. SK2 inhibition should, theoretically, decrease sphingosine-1-phosphate and increase ceramide levels—recent studies, however, suggest that some SK2 inhibitors, such as the one that both we and others have shown to inhibit CHIKV, ABC294640, have the opposite effect, increasing S-1-P through unknown mechanisms [106]. While not necessarily a shift from glycosylated ceramides to ceramides, Zika infection was associated with increased sphingomyelin to ceramide flux; however, it had the opposite effect to that seen in CHIKV infection: increased ceramide levels were advantageous to Zika replication. As we see in Figure 6, there is no unified response to an increase in ceramides or a decrease in biosynthesis via SPT1 inhibition.

Among the hexosylceramide anabolic enzymes, inhibition of glucosylceramide synthase (GCS) and UDP-galactosyltransferase (UGT8) will both increase ceramides. However, only inhibition of galactosylceramide synthesis via UGT8-Inh-19 has an effect, one which reduces CHIKV entry, replication, and infectious virus production. How galactosyceramide is important for CHIKV entry is unclear: the vast majority of research surrounding GalCer pertains to its connection to the myelin-sheath associated sulfur-modified GalCer, sulfatide [107,108], and furthermore recent studies suggest that UGT8 does not only catalyze glycosylation of GalCer, but also the synthesis of monogalactosyldiacylglycerol (MGDG), a minor membrane lipid [109]. Galactosylceramide and sulfatide [54] are known to be present in lipid rafts and to increase their thickness, shield the resident cholesterol from water, and slow lateral diffusion [54,110]. Depletion of GalCer or sulfatide could impair the known association of the CHIKV replication complex with lipid rafts [21]. Further studies could look into the effect of blocking the degradation of hexosylceramides, especially galactosylceramides, on CHIKV infection. We predict that blocking the degradation of HexCer might induce greater permissivity of CHIKV within the cell, indicating that the increased ceramide/hexosylceramide flux in infected cells may well be an antiviral response.

Elucidation of the mechanism of CHIKV sensitivity to CETP inhibitors, specifically evacetrapib, is unclear. The relevance of CETP within cell culture should be exclusive to cells grown in media supplemented with FBS, which contains apolipoprotein B (ApoB) LDL, apolipoprotein A-I (ApoA-I) HDL, and minor amounts of VLDL [111]. CETP converts HDL to atherosclerotic LDL. CETP inhibition, consequently, increases HDL levels and decreases LDL conversion, hence the pursuit of inhibitors as a pharmaceutical to treat cardiovascular disease. A potential mechanism, thus, would be that HDL conversion and total LDL levels are proviral factors that are inhibited by CETP inhibition. This mechanism is supported by the fact that the majority of cellular cholesterol is obtained via LDL:LDLR-mediated endocytosis [112]. However, we observed that evacetrapib continued to block CHIKV entry even in the absence of serum proteins, suggesting an off-target mechanism of inhibition. Prior studies of evacetrapib found no off-target effects that have been evident in other CETP inhibitors, suggesting that there are hitherto unknown cellular targets [113,114].

The findings of this study must be interpreted in light of certain limitations. Vero cells, while being a commonly used cell line in biomedical research and basic virology of human pathogens, including CHIKV, are derived from African Green monkey kidney tissue [115,116]. While nephritis has been observed in a handful of cases, kidneys are not noted to be a major site of CHIKV replication in in vivo infection [89,117]. Vero cells cannot produce interferon and lack cyclin-dependent kinase inhibition [118]. Without these important immune and cell regulatory mechanisms, it is possible that the results seen in this study may not be observed in a primary human cell model of CHIKV infection. Thus, further studies should aim to characterize whether the hexosylceramide flux, dependence on galactosylceramide synthesis, and inhibitory effect of evacetrapib are present in more applicable primary cell lines, including monocytic cell lines and liver tissues, all of which are infected and play an important role in human disease [90,119,120]. In addition to mammalian-derived cell models of CHIKV infection, future studies should aim to characterize how CHIKV infection affects the unique cellular lipidome of its other host, mosquitoes [121,122]. The use of small-molecule inhibitors and agonists throughout this study, too, poses a limitation. Many small molecules can have off-target effects [123,124,125]. Future studies should utilize genetic knockdowns or knock-outs of enzymes responsible for hexosylceramide metabolism to confirm the role of these lipids in CHIKV replication.

Our results demonstrate that CHIKV infection induces targeted alterations in host cell lipids—most notably, reducing hexosylceramides while increasing ceramides. Unlike other positive-sense RNA viruses, CHIKV did not broadly remodel phospholipid abundance and was dependent on specific lipid metabolic pathways. Blocking fatty acid synthesis and cholesterol pathways inhibited CHIKV replication, as has been observed by other groups, but we also observed that galactosylceramide biosynthesis is critical for CHIKV, and it is sensitive to evacetrapib through an unknown mechanism. These findings highlight previously underappreciated metabolic vulnerabilities in the CHIKV life cycle and point toward lipid-targeted strategies as potential antiviral interventions.

## Figures and Tables

**Figure 1 viruses-17-00509-f001:**
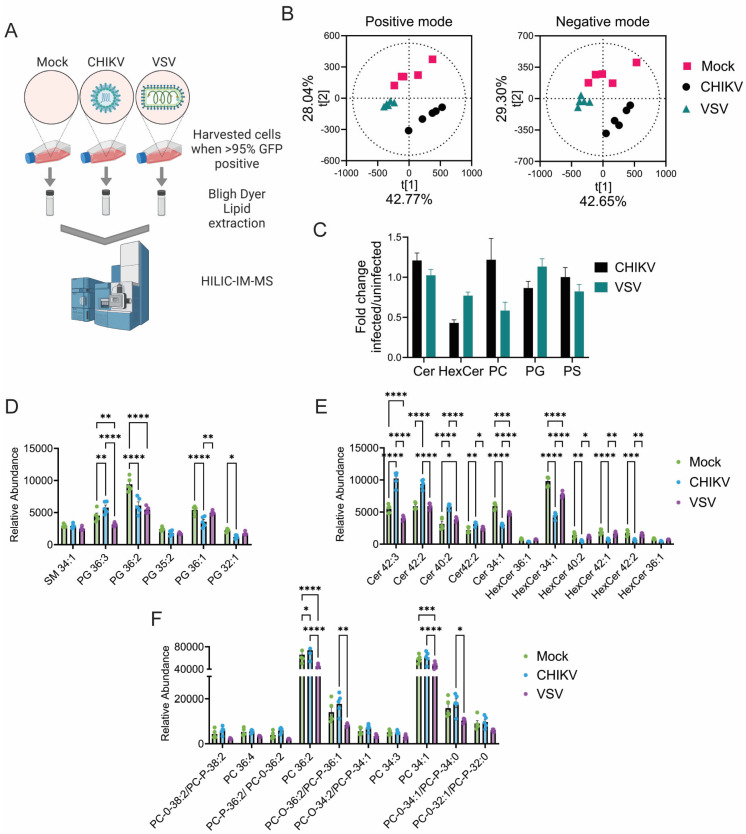
CHIKV infection decreases HexCer levels. (**A**) Schematic of the lipidomics experiment. Created in biorender.com. (**B**) PCA analysis of the lipidomic samples in both the positive and negative ion modes. (**C**) Fold change in levels of lipids found in chikungunya (CHIKV) and vesicular stomatitis virus (VSV) infected cells compared to uninfected cells. (**D**–**F**) Relative abundances of lipids driving the PCA analysis. Ceramide (Cer), hexosylceramides (HexCer), phosphatidylcholine (PC), phosphatidylglycerol (PG), and phosphatidylserine (PS). Two-way ANOVA with post hoc Tukey’s multiple comparisons test. *, *p* ≤ 0.05, **, *p* ≤ 0.01, ***, *p* ≤ 0.001, ****, *p* ≤ 0.0001.

**Figure 2 viruses-17-00509-f002:**
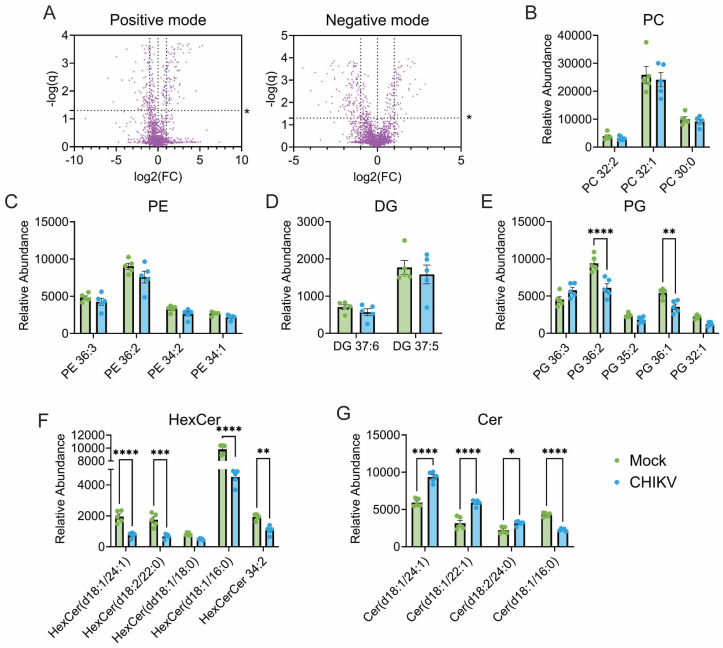
CHIKV infection-associated cellular lipidome. (**A**) Volcano plots of untargeted mass-spec lipidomic datasets in CHIKV-infected cells compared to uninfected controls in both positive (**Left**) and negative (**Right**) ion modes. (**B**–**G**) Bar charts of lipid species identified via multivariate analysis to contribute to significant variance between CHIKV-infected and mock-infected lipidomes. Phosphatidylcholine (PC), phosphatidylethanolamine (PE), diacylglycerol (DG), phosphatidylglycerol (PG), hexosylceramide (HexCer), ceramide (Cer). Data are based on the averages of *n* = 5 replicates. For (**A**), significance was determined via multiple unpaired *t*-tests (Benjamini, Krieger, Yekutieli, FDR = 1.0%). In (**B**–**G**), Two-way ANOVA with post hoc Sidak’s multiple comparisons test was employed for univariate analysis. *, *p* ≤ 0.05, **, *p* ≤ 0.01, ***, *p* ≤ 0.001, ****, *p* ≤ 0.0001.

**Figure 3 viruses-17-00509-f003:**
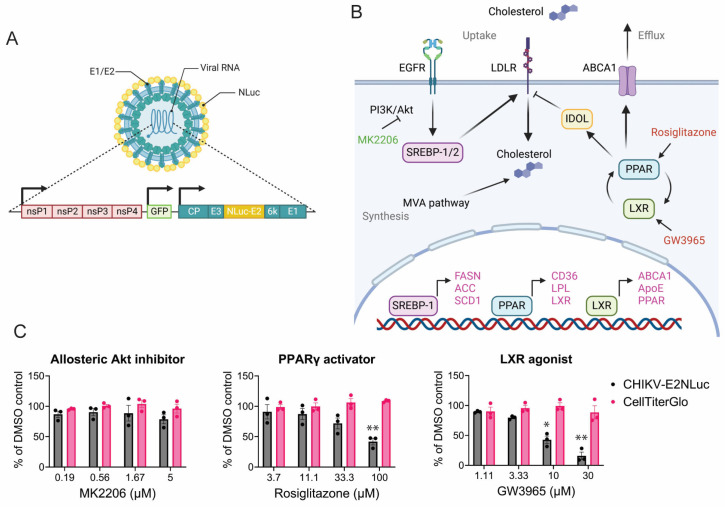
Altering lipid regulatory pathways reduced CHIKV replication. (**A**) CHIKV virus contains both a GFP and nano-luciferase (NLuc) to readily monitor virus replication. (**B**) Pathways that can lead to transcriptional regulation of lipid regulatory pathways. (**C**) Relative CHIKV replication and cell viability with MK2206, rosiglitazone, and GW3965. Data represent the average of triplicates completed in three independent trials compared to the DMSO control. An unequal variance (Welch’s correction) *t*-test was performed for normalized data. *, *p* < 0.05; **, *p* < 0.01. (**A**,**B**) created with biorender.com.

**Figure 4 viruses-17-00509-f004:**
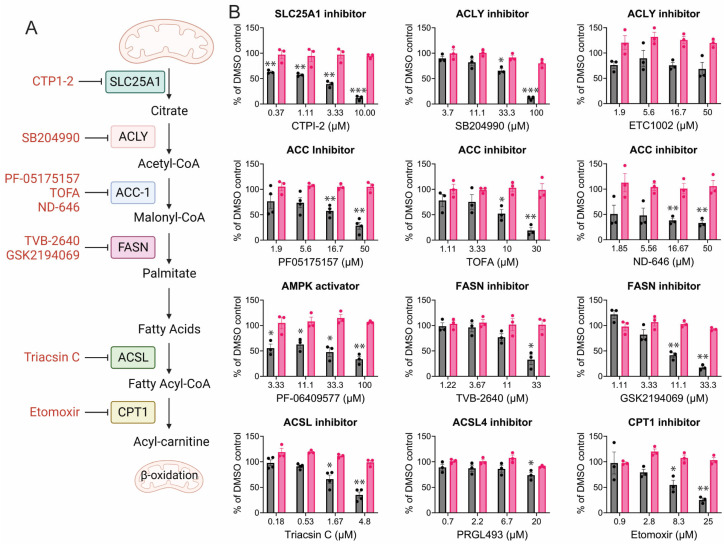
CHIKV replication is sensitive to disruption of cellular fatty acid synthesis. (**A**) Pathway describing fatty acid synthesis, activation, and catabolism, created in biorender.com. (**B**) CHIKV replication (dark grey) and cell viability (pink) when infection occurs in the presence of the indicated inhibitor. Data represent the average of triplicates completed in at least three independent trials compared to the DMSO control. An unequal variance (Welch’s correction) *t*-test was performed for normalized data. *, *p* < 0.05; **, *p* < 0.01; ***, *p* < 0.001.

**Figure 5 viruses-17-00509-f005:**
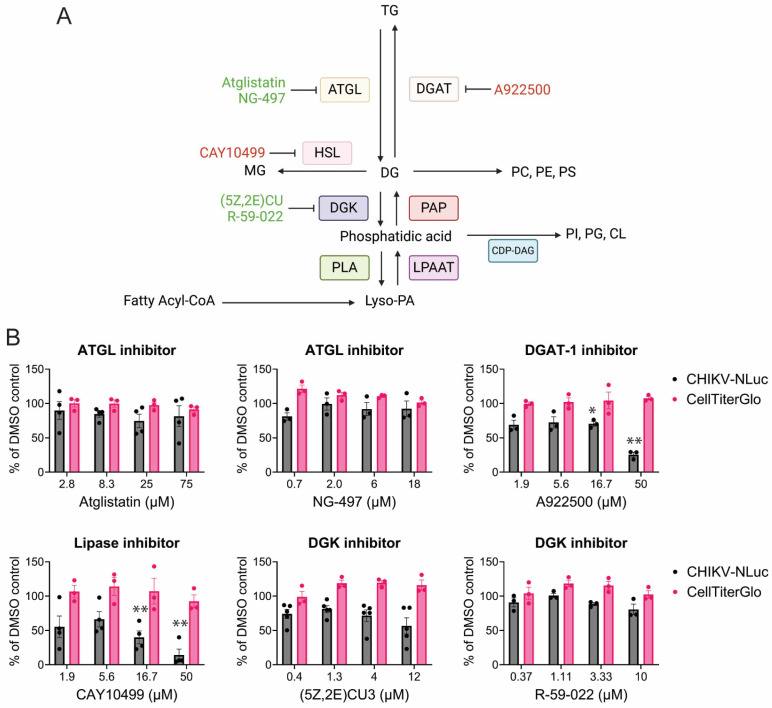
CHIKV replication is sensitive to lipogenesis and lipolysis inhibitors. (**A**) Phospholipid to triacylglyceride anabolic and catabolic pathway, with complex phospholipid biosynthesis branches, created in biorender.com. Lyso-phosphatidic acid (Lyso-PA), monoacyglycerol (MAG), phosphatidylinositol (PI), cardiolipin (CL) (**B**) CHIKV replication (dark grey), and cell viability (pink) when infection occurs in the presence of the indicated inhibitor. Data represent the average of triplicates completed in at least three independent trials compared to the DMSO control. An unequal variance (Welch’s correction) *t*-test was performed for normalized data. *, *p* < 0.05; **, *p* < 0.01.

**Figure 6 viruses-17-00509-f006:**
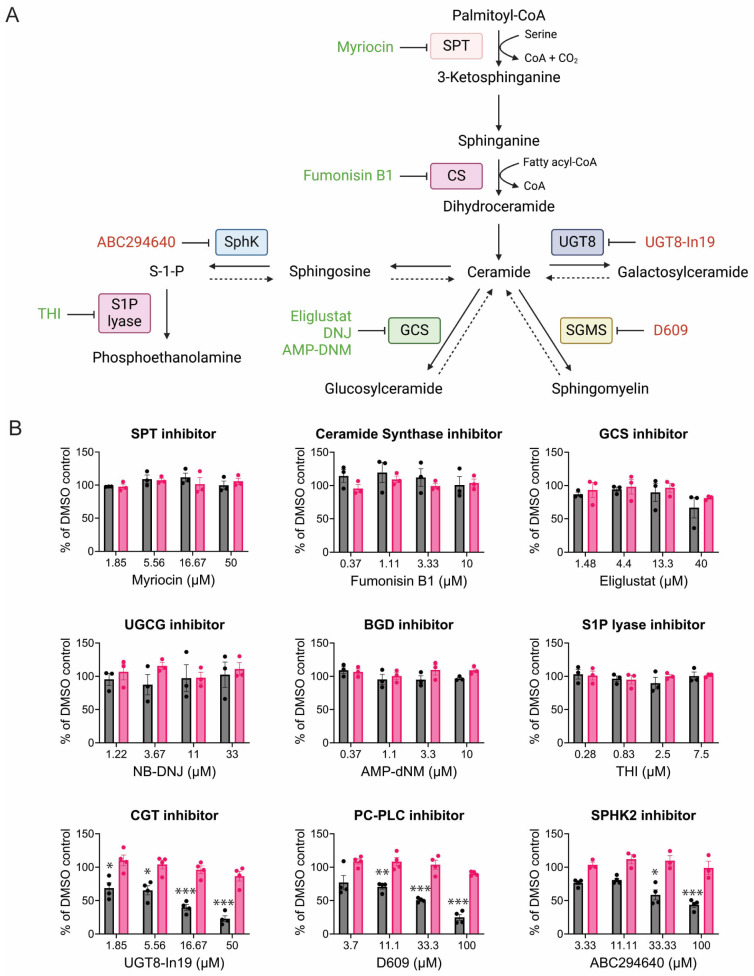
CHIKV replication and ceramide synthesis. (**A**) Ceramide and sphingolipid synthesis pathway, created in biorender.com. (**B**) CHIKV replication (dark grey) and cell viability (pink) when infection occurs in the presence of the indicated inhibitor. Data represent the average of triplicates completed in at least three independent trials compared to the DMSO control. An unequal variance (Welch’s correction) *t*-test was performed for normalized data. *, *p* < 0.05; **, *p* < 0.01; ***, *p* < 0.001.

**Figure 7 viruses-17-00509-f007:**
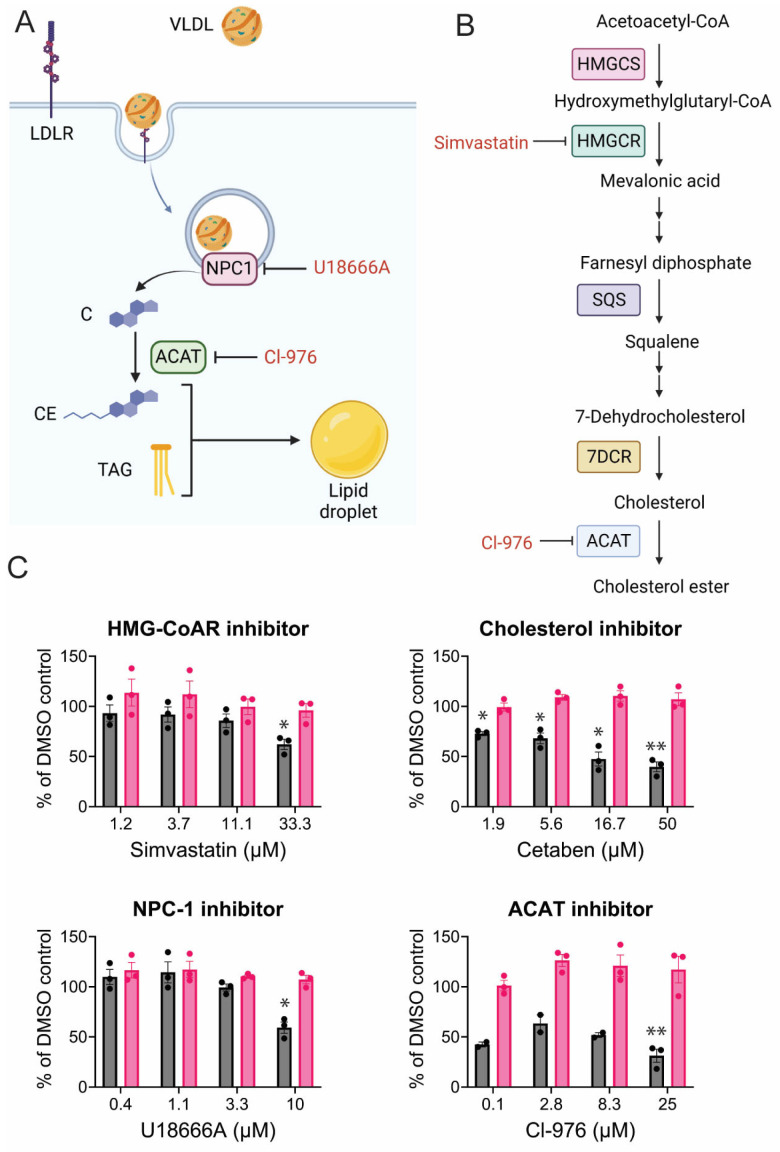
CHIKV and cholesterol. (**A**) Cholesterol uptake pathway and (**B**) cholesterol synthesis pathways; created in biorender.com. (**C**) CHIKV replication (dark grey) and cell viability (pink) when infection occurs in the presence of the indicated inhibitor. Data represent the average of triplicates completed in at least three independent trials compared to the DMSO control. An unequal variance (Welch’s correction) *t*-test was performed for normalized data. *, *p* < 0.05; **, *p* < 0.01.

**Figure 8 viruses-17-00509-f008:**
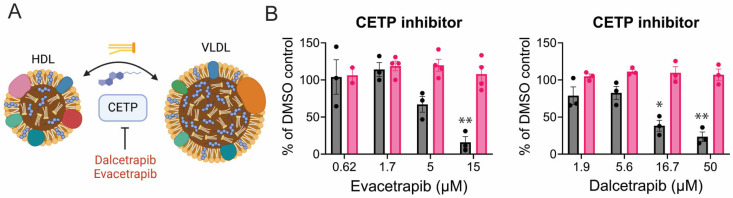
CETP inhibitors limit CHIKV replication. (**A**) Schematic of CETP function; created in biorender.com. (**B**) CHIKV replication (dark grey) and cell viability (pink) when infection occurs in the presence of the indicated inhibitor. Data represent the average of triplicates completed in at least three independent trials compared to the DMSO control. An unequal variance (Welch’s correction) *t*-test was performed for normalized data. *, *p* < 0.05; **, *p* < 0.01.

**Figure 9 viruses-17-00509-f009:**
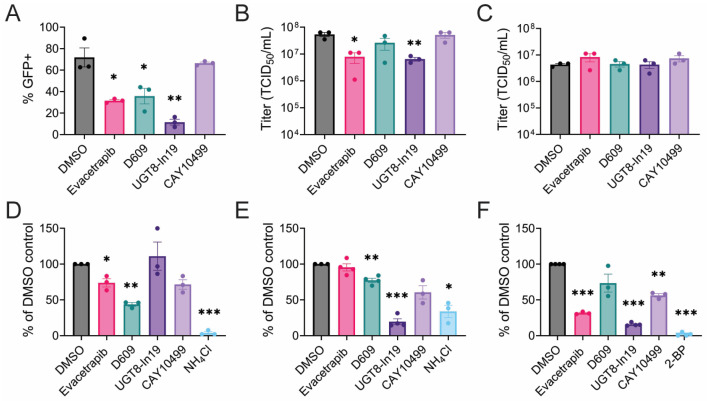
Mechanism of inhibition. (**A**) CHIKV spread over 24 h, MOI = 0.001. (**B**) Supernatant titers after 24 h treatment, MOI = 0.001. (**C**) Virucidal assay. Viral stocks were incubated with compounds for 1 h, then titrated. (**D**) Virus entry after short, 1 h compound pre-treatment. (**E**) Virus entry after long, 18 h compound pre-treatment. (**F**) CHIKV replicon assay. When listed, compounds were at the following concentrations: DMSO [0.1% *v*/*v*], evacetrapib [15 µM], D609 [100 µM], UGT8-In19 [25 µM], CAY10499 [50 µM], NH_4_Cl [10 mM], 2-BP [50 µM]. Data represent the mean ± SEM from at least three independent trials. Student *t*-test was used to compare treated samples with DMSO control. An unequal variance (Welch’s correction) *t*-test was performed for normalized data. *, *p* < 0.05; **, *p* < 0.01; ***, *p* < 0.001.

**Figure 10 viruses-17-00509-f010:**
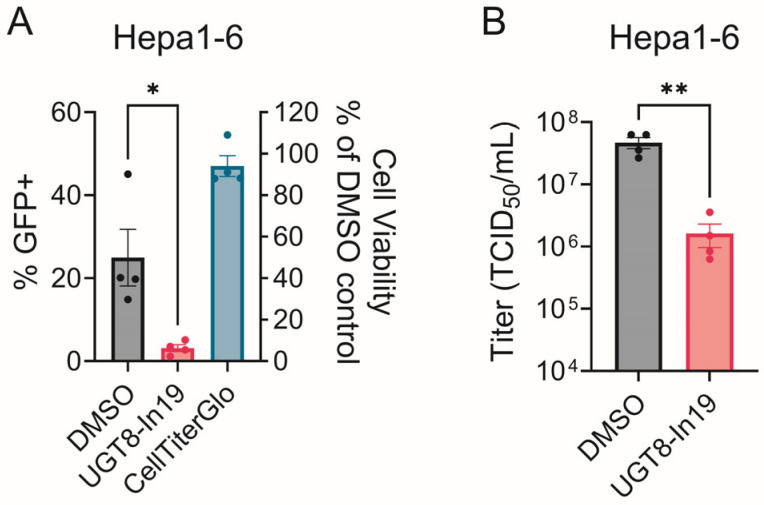
UGT8-In19 inhibits CHIKV in Hepa1-6 cells. (**A**) CHIKV spread occurred over 48 h in Hepa1-6 cells, MOI = 0.001. Cell viability of uninfected UGT8-In19 [8.3 µM] treated cells is shown compared to DMSO control. (**B**) Supernatant titers after 48 h treatment, MOI = 0.01. Data represent the mean ± SEM from at least three independent trials. Student *t*-test was used to compare treated samples with DMSO control. An unequal variance (Welch’s correction) *t*-test was performed for normalized data. *, *p* < 0.05; **, *p* < 0.01.

**Figure 11 viruses-17-00509-f011:**
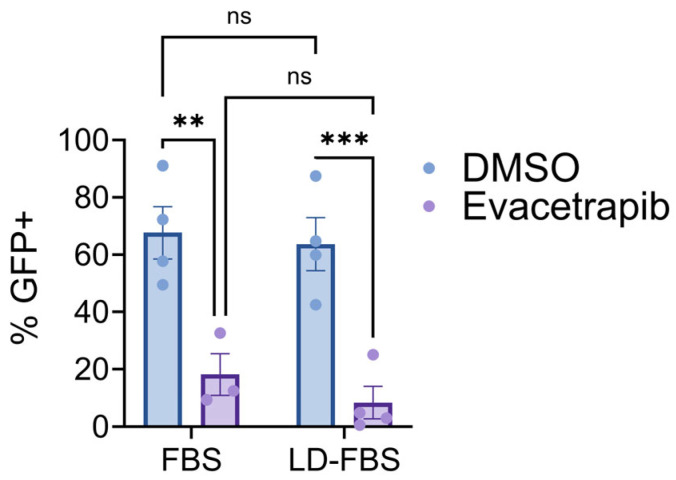
Evacetrapib inhibition is independent of serum lipoproteins. VeroS cells were infected with CHIKV (MOI = 0.001) and treated with Evacetrapib or DMSO within DMEM-media containing normal FBS or lipoprotein-depleted FBS (LD-FBS); cells were harvested at 24 h and percent infection determined via flow cytometry. Two-way ANOVA with post hoc Sidak’s multiple comparison test was employed for statistical analysis. **, *p* ≤ 0.01, ***, *p* ≤ 0.001.

## Data Availability

Raw lipidomics data files and processed result files are available on the MassIVE repository (MSV000096887).

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
