# Peer review of "Chikungunya Replication and Infection Is Dependent upon and Alters Cellular Hexosylceramide Levels in Vero Cells"

_viruses, 2025, doi:10.3390/v17040509_

Round 1
Reviewer 1 Report
Comments and Suggestions for Authors
This study by Noble et al. investigates the crosstalk between CHIKV and mammalian host lipid metabolism. Through an untargeted lipidomic analysis of Vero cells infected with CHIKV, the authors report virus-induced alterations in sphingolipid metabolism, including the catabolism of hexosylceramides and an increased availability of ceramides. The manuscript also presents inhibitor-based studies aimed at investigating the functional impact of fatty acids, neutral lipids, sphingolipids, and cholesterol biosynthesis pathways on CHIKV infection. The study identifies specific steps in the CHIKV replication cycle that are affected by these lipid pathways, supporting a direct requirement for galactosylceramides in virus entry and genome replication. This research provides valuable insights into how CHIKV manipulates host lipid biosynthesis for optimal replication, though these findings should be validated in a more relevant human cell model.
Main Points:
The entire study is conducted using Vero and Vero S cells, both derived from monkey kidney. How relevant are these cell lines for studying the CHIKV/human cell metabolism crosstalk? While Vero cells are commonly used in viral infection studies, they may not fully mimic the lipidomic response of human cells to CHIKV infection. Additionally, Vero cells lack interferon receptors, meaning they cannot activate key innate immune response pathways, such as the interferon pathway. This could influence lipid alterations observed during infection, especially those involved in antiviral defense. The rationale for using Vero cells expressing the human signaling lymphocytic molecule (SLAM) is unclear. Therefore, the functional aspect of the study should include controls using a human cell model more closely relevant to the CHIKV infection process.
The lipidomic data should be presented in greater detail. Figures 1 and 2 focus only on selected lipid classes (sphingolipids, phospholipids, and neutral lipids). However, how are lipid species from metabolic pathways considered in the functional analysis regulated during CHIKV infection? Also, why do the lipids presented in Figures 1 and 2 differ?
Functional studies primarily rely on chemical inhibitors, which can have off-target effects and disrupt multiple cellular pathways beyond the intended target. For instance, rostiglitazone inhibits lipid and energy metabolism but also reduces NF-kB activation and proinflammatory responses. How do the authors distinguish between these effects? Additionally, using genetic invalidation approaches could help clarify the specificity of evacetrapib and dalcetrapib’s antiviral activities. The mechanisms of action for ETC102, PF-06409577, PRGL493, Cetaben, and Etomoxir should be clarified in Figures 4 and 7.
The results appear somewhat contradictory between the figures. For example, how do the authors explain that 50 µM CAY 10499 dramatically reduces CHIKV infection in Figure 5, but decreases the efficacy of virus entry and genome replication (Figures 9D-F), without reducing the percentage of CHIKV-positive cells or viral titers in Figure 9?
Minor Points:
- Add the Addgene reference for the CHIKV replicon used in the study.
- The last two figures in the manuscript are numbered as Figure 9F.
- In the experimental section, the infectious dose should be presented consistently (MOI or TCID50).
- Make the nomenclature consistent for diacylglycerols: DG in Figure 2 and DAG in Figure 5.
- In Figure 7, change “Simvistatin” to Simvastatin.
- In the manuscript, the authors refer to MK-2206 as an inhibitor of SREBP2, while Figure 3 depicts it as a SREBP1 inhibitor. Please clarify.
Author Response
Reviewer 1: This study by Noble et al. investigates the crosstalk between CHIKV and mammalian host lipid metabolism. Through an untargeted lipidomic analysis of Vero cells infected with CHIKV, the authors report virus-induced alterations in sphingolipid metabolism, including the catabolism of hexosylceramides and an increased availability of ceramides. The manuscript also presents inhibitor-based studies aimed at investigating the functional impact of fatty acids, neutral lipids, sphingolipids, and cholesterol biosynthesis pathways on CHIKV infection. The study identifies specific steps in the CHIKV replication cycle that are affected by these lipid pathways, supporting a direct requirement for galactosylceramides in virus entry and genome replication. This research provides valuable insights into how CHIKV manipulates host lipid biosynthesis for optimal replication, though these findings should be validated in a more relevant human cell model.
Main Points:
The entire study is conducted using Vero and Vero S cells, both derived from monkey kidney. How relevant are these cell lines for studying the CHIKV/human cell metabolism crosstalk? While Vero cells are commonly used in viral infection studies, they may not fully mimic the lipidomic response of human cells to CHIKV infection. Additionally, Vero cells lack interferon receptors, meaning they cannot activate key innate immune response pathways, such as the interferon pathway. This could influence lipid alterations observed during infection, especially those involved in antiviral defense. The rationale for using Vero cells expressing the human signaling lymphocytic molecule (SLAM) is unclear. Therefore, the functional aspect of the study should include controls using a human cell model more closely relevant to the CHIKV infection process.
Author Response: Vero cells are commonly used in virus stock production because they lack the interferon response. While the cells may not represent a disease relevant cell type, they are readily infectable (both susceptible and permissive) and are commonly used to produce virus stocks. CHIKV entry in Vero cells is associated with phosphatidylserine receptors (PMID: 34359995; PMID: 36743418; PMID: 39007616). We have also used Vero cells to monitor VSV induced lipid changes (PMID: 35062207), which helped us compare the virus specific changes that occur in one cells type. We used two different Vero lines to increase the rigor of our findings. The rationale behind using VeroS cells, especially in Fig 9, was an increase in our observed transfection efficiency relative to our VeroE6 cells. An effective transfection efficiency was critical for the replicon experiment (Fig 9F) and the rest of Fig 9 is performed in VeroS cells so that all results are within the same cell type. This paper serves as a base for further studies regarding CHIKV sensitivity to galactosylceramide synthesis in more applicable cell lines (THP-1 derived macrophages and primary cell lines). We have included additional discussion about the limitations to using Vero cells in the discussion (line 403-420).
Reviewer 1: The lipidomic data should be presented in greater detail. Figures 1 and 2 focus only on selected lipid classes (sphingolipids, phospholipids, and neutral lipids). However, how are lipid species from metabolic pathways considered in the functional analysis regulated during CHIKV infection? Also, why do the lipids presented in Figures 1 and 2 differ?
Author Response: It is explained in the text that Fig 1 reports on the significant differences between VSV and CHIKV, and that there are independent cellular lipidomic responses to infection with each virus. The bar charts for lipidomic data in Fig 1 reports solely on species indicated to be of differential abundance by PCA analysis between mock cells and those infected with VSV vs CHIKV. We focus on the major differences between lipid class in this figure to demonstrate the observed changes were different when comparing different viruses. Figure 2, on the other hand, reports the significant difference between mock and CHIKV, thus not every species of lipid reported in Fig 1 will be reported in Fig 2. In figure 2 we are more detailed and show the specific lipid species differences between mock and CHIKV infected cells.
Reviewer 1: Functional studies primarily rely on chemical inhibitors, which can have off-target effects and disrupt multiple cellular pathways beyond the intended target. For instance, rostiglitazone inhibits lipid and energy metabolism but also reduces NF-kB activation and proinflammatory responses. How do the authors distinguish between these effects? Additionally, using genetic invalidation approaches could help clarify the specificity of evacetrapib and dalcetrapib’s antiviral activities. The mechanisms of action for ETC102, PF-06409577, PRGL493, Cetaben, and Etomoxir should be clarified in Figures 4 and 7.
Author Response: Thank you for pointing these multiple points out. Regarding rosiglitazone, we do not distinguish between the effects of metabolism and NF-KB, and merely state that rosiglitazone, a PPARY agonist associated with lipid regulation, affects virus (PMID 26047949). For the lipid regulatory element paragraph, we have noted that the responses of CHIKV of LRE inhibitors cannot be solely attributed to their affect on lipid regulation, as SREBP, LXR, and PPARY are associated with many other pathways.
Regarding dalcetrapib and evacetrapib antiviral activities, we agree with your comment, and add that cellular thermal shift assay might also aid in identifying the target of these drugs. We have included mention of genetic assays in the discussion (lines )
The mechanism of action for many of the inhibitors of fatty acid synthesis, as well as for the mentioned ones ETC1002, PF-06409577, PRGL43, cetaben, and etomoxir have been clarified.
Reviewer 1: The results appear somewhat contradictory between the figures. For example, how do the authors explain that 50 µM CAY 10499 dramatically reduces CHIKV infection in Figure 5, but decreases the efficacy of virus entry and genome replication (Figures 9D-F), without reducing the percentage of CHIKV-positive cells or viral titers in Figure 9?
Author Response: Regarding figure 9D, no statistically significant effect on entry was observed with CAY10499 treatment. Furthermore, while the effect on replicon reporter gene was significant (Fig9F), it should be noted that 1) a 50% effect on replicon reporter gene levels, while statistically significant, may not be enough to drop noticeably drop titer levels in a multistep infection (Fig 9B). The drugs which did have an effect on titer levels, evacetrapib and ugt8-IN-19, each dropped replicon level roughly 90%, corresponding to a log decrease in both replicon levels and titers. Given that 9B is a multi-step infection (.001MOI, 48hrs), it is possible that the small effect of CAY on replication is masked by multiple steps of viral infection
The discrepancy between the results of Fig 5 and Fig 9 suggest that inhibitors that show a strong CHIKV-inhibitory effect in Vero cells may not necessarily share an effect to the same degree in VeroS cells. As mentioned before, VeroS cells were employed in Fig 9 due to their higher transfection efficiency.
Minor Points:
-Reviewer 1: Add the Addgene reference for the CHIKV replicon used in the study.
Author: The Addgene accession number for Pchikv-NlucP-eGFP, #232249, has been added. (LINE 556)
-Reviewer 1: The last two figures in the manuscript are numbered as Figure 9F.
Author: Corrected the last figure to “Figure 10”. (LINE 313)
-Reviewer 1: In the experimental section, the infectious dose should be presented consistently (MOI or TCID50).
Author: The drug inhibition assay has been corrected from “300 particles per well” to the corresponding MOI, 0.003 MOI. (LINE 504)
-Reviewer 1: Make the nomenclature consistent for diacylglycerols: DG in Figure 2 and DAG in Figure 5.
Author: The nomenclature for TG, DG, and MG have been adjusted to align with LIPIDMAPs terminology and is consistent throughout the paper.
-Reviewer 1: In Figure 7, change “Simvistatin” to Simvastatin.
Author: Figure 7 has been adjusted to the the correct spelling, “simvastatin”. (PAGE 10)
-Reviewer 1: In the manuscript, the authors refer to MK-2206 as an inhibitor of SREBP2, while Figure 3 depicts it as a SREBP1 inhibitor. Please clarify.
Author: The figure has been adjusted to show that MK2206 inhibits Akt, which is upstream of SREBP1/2, and the text has been adjusted to indicate that this is an indirect action caused by pi3k/akt/mtor inhibition.
Reviewer 2 Report
Comments and Suggestions for Authors
For authors
Some following spelling errors were found for improvements:
Line 241: change „figure 8B” with “Figure 8B”;
Lines 269, 304 and 473 insert space characters in appropriate places.
Line 461: check the spelling of “100ul" and replace it with “100 μL”,
Line 472: correct spelling of “μl”: with “μL”.

Author Response
Thank you for your careful review. We have corrected all of the listed spelling and formatting errors.
Reviewer 3 Report
Comments and Suggestions for Authors
The article without a doubt presents interesting topics and approaches. It's clear, well-modulated, and scientifically quite strong.
However, I would like to highlight a few points:
1 The use of Vero cells has some strong limitations, and the authors should mention it. There is a lack of understanding of virus-host interactions during infection and cell-based virus production in Vero cells.
2 In addition, some results have shown that chikungunya virus (CHIKV) could be replicated better in tissue-engineered biological models than in cell culture. The authors should mention this.
3 An essential point that should be also mentioned in a separate paragraph is the type of infected subject. The elderly and newborns are more at risk of developing long-term complications from chikungunya. People with pre-existing health conditions are also at risk of developing more serious long-term effects such as diabetes, heart disease, and hypertension.
4 Please consider in this specific case the cytopathic effects of Vero cells that may alter the cholesterol integrity due to the vacuolation and formation of syncytia as a result of apoptotic mechanism
Author Response
1 The use of Vero cells has some strong limitations, and the authors should mention it. There is a lack of understanding of virus-host interactions during infection and cell-based virus production in Vero cells.
We have acknowledged the limitations of Vero cells in the penultimate paragraph within our discussion (lines 405-420). The interaction of Vero cells with viruses has been very well characterized, but we note that the results might not directly translate to CHIKV infection of more relevant cell types.
2 In addition, some results have shown that chikungunya virus (CHIKV) could be replicated better in tissue-engineered biological models than in cell culture. The authors should mention this.
That is an excellent point. While it is outside the scope of this study, we do include in our penultimate paragraph that we intend to perform future cell lines or model systems. (405-420)
3 An essential point that should be also mentioned in a separate paragraph is the type of infected subject. The elderly and newborns are more at risk of developing long-term complications from chikungunya. People with pre-existing health conditions are also at risk of developing more serious long-term effects such as diabetes, heart disease, and hypertension.
A sentence has been added to the introduction that describes the differential response to CHIKV that has been observed in immunocompromised individuals. (34-36)
4 Please consider in this specific case the cytopathic effects of Vero cells that may alter the cholesterol integrity due to the vacuolation and formation of syncytia as a result of apoptotic mechanism
The results from the lipidomics, as well as all other experiments, were obtained prior to virus induced cytopathic effect and subsequent apoptosis (within 24hrs of infection). Virus-induced syncytia are not observed within CHIKV infected Vero at the physiologic pH maintained in cell culture.
Reviewer 4 Report
Comments and Suggestions for Authors
In this article by Noble et al entitled manuscript “Chikungunya replication and infection is dependent upon and alters cellular hexosylceramide levels in Vero cells” how CHIKV targets the whole lipid metabolism during infection and identified glycosphingolipid metabolism to be an important pathway through functional analysis as well as inhibitor assays. The article is well written, however the data representation needs a mend.
There are few areas that needs further clarification.
(i) Since the lipidomics data after CHIKV infection hasn’t been done before, it would be good to provide more comprehensive data done with HILIC-IM-IS. It would be ideal to show more data and functional studies from data set of HILIC-IM-IS Instead of figure 1C, the authors can represent the amount of each types of lipid levels in different conditions like Mock, CHIKV infected and VSV infected in a pie chart or other appropriate representation with proper statistics. Also, for those shown the relative abundance differences (statistically significant, figure 2B-G) need a confirmation with other assays.
(ii) The graphical representation of viral titres together with cell viability in the current representation is not ideal. It’s always good to compare the effect of drug with its vehicle control and represent in such a way to show the statistical significance similar to figure 9. So my recommendation is either group the cell viability/viral titer conditions together, so that you can compare among the concentrations, or compare with the vehicle controls.
(iii) Justify why the Vero Slam cells are used for infection.
Author Response
Rewiever Comment 1: Since the lipidomics data after CHIKV infection hasn’t been done before, it would be good to provide more comprehensive data done with HILIC-IM-IS. It would be ideal to show more data and functional studies from data set of HILIC-IM-IS Instead of figure 1C, the authors can represent the amount of each types of lipid levels in different conditions like Mock, CHIKV infected and VSV infected in a pie chart or other appropriate representation with proper statistics. Also, for those shown the relative abundance differences (statistically significant, figure 2B-G) need a confirmation with other assays.
Author answer 1: We thank Reviewer 4 for their excellent critique. To represent a more comprehensive picture of the lipidomic response of Vero cells to mock, VSV, and CHIKV infection, we have added to Figure 1 charts reflecting the lipidomic data in each group with stats
Rewiever Comment 2: The graphical representation of viral titres together with cell viability in the current representation is not ideal. It’s always good to compare the effect of drug with its vehicle control and represent in such a way to show the statistical significance similar to figure 9. So my recommendation is either group the cell viability/viral titer conditions together, so that you can compare among the concentrations, or compare with the vehicle controls.
Author answer 2: It is our understanding that Reviewer 4 is referencing graphs within Figures 3-8. It should be noted, as stated in the text and in figure 3, that viral titers are not being measured, but luciferase expression from a nano-luciferase expressing recombinant CHIKV, as a measure of viral replication. Furthermore, Figures 3-8 represent both cytotoxicity (Gray) and luciferase level (Pink) as a % of vehicle (DMSO) control, as requested by the reviewer. Additionally, statistics have been applied to figures 3-8, as suggested by the reviewer; where significant, a corresponding number of asterisks have been applied.
Reviewer Comment 3: Justify why the Vero Slam cells are used for infection.
Author answer 3: There are multiple rationales behind using VeroS cells within this study. Firstly, the Brindley & Hines lab has published previous lipidomic studies with VSV infection of VeroS cells (Havranek et al 2021), allowing us to compare our CHIKV findings with previous studies. Furthermore, previous studies have established that VeroE6 and VeroS establish equal responses to viral infection of CHIKV (Reyes Ballista et al 2023) The specific rationale for the implementation of VeroS cells in Fig 9 was an increase in our observed transfection efficiency relative to our VeroE6 cells. An effective transfection efficiency was critical for the replicon experiment (Fig 9F) and the rest of Fig 9 is performed in VeroS cells so that all results are within the same cell type.
Round 2
Reviewer 1 Report
Comments and Suggestions for Authors
The manuscript has been improved, particularly by including a thoughtful discussion on the limitations of the study related to the cell model and the use of drugs with well-documented off-target effects in the literature. However, some additional key points can easily addressed.
Regarding the lack of access to the complete lipidomic data in Figure 1, the choice to focus on specific lipid species showing significant differences between the mock and CHIKV conditions is well argued in the original manuscript. However, it would be beneficial to include these additional data in the figure to provide a more comprehensive view of the findings.
While the authors' acknowledgment of the limitations of using Vero cells (well known for virus stock production) is appreciated, validation of key results in a cell model more relevant to CHIKV infection in humans is still necessary. This is especially important since the effects of the drugs tested may vary depending on the cell type, as emphasized by the authors in their rebuttal letter comparing results in Vero and VeroS cells presented in Figures 5 and 9.
Author Response
In regards to the comment: Regarding the lack of access to the complete lipidomic data in Figure 1, the choice to focus on specific lipid species showing significant differences between the mock and CHIKV conditions is well argued in the original manuscript. However, it would be beneficial to include these additional data in the figure to provide a more comprehensive view of the findings.
The lipidomics data shown in figures 1 and 2 are both showing results from the same experiment in different ways. In figure 1, we focus on the changes in lipid species that changed the most when comparing uninfected to CHIKV and uninfected to VSV. This is primarily to demonstrated the lipid changes we are seeing are unique to CHIKV and not simply a pattern observed after any viral infection. We published the specific lipidomics results from VSV infected Vero cells (PMID: 35062207) in 2021. Figure 2 shows the comprehensive lipid changes that occur from CHIKV infection in detail. We are unclear what data we should show to "provide a more comprehensive view of the findings".
In regards to the comment "While the authors' acknowledgment of the limitations of using Vero cells (well known for virus stock production) is appreciated, validation of key results in a cell model more relevant to CHIKV infection in humans is still necessary. This is especially important since the effects of the drugs tested may vary depending on the cell type, as emphasized by the authors in their rebuttal letter comparing results in Vero and VeroS cells presented in Figures 5 and 9."
We are very careful throughout the manuscript to conclude that the results are related to Vero cells, hence the title, etc. While demonstrating the drugs work in human cell lines would be relevant if we were suggesting these drugs could be used for treatment, we are not. This study was to explore how changes in lipid pathways alter replication in Vero cells and nothing more. Future studies can expand studies to determine if these pathways are relevant in disease related tissues or animal models of disease, but is outside the scope of this study.
Reviewer 3 Report
Comments and Suggestions for Authors
The manuscript can be accepted
Author Response
Thank you!
Round 3
Reviewer 1 Report
Comments and Suggestions for Authors
Noble et al.'s response regarding manuscript “Chikungunya replication and infection is dependent upon and alters cellular hexosylceramide levels in Vero cells” focuses on two main points: (i) the availability of untargeted lipidomics data in the context of CHIKV infection and (ii) the relevance of the cellular model used for the study.
As detailed by the authors, the manuscript examines variations in selected lipid species (Figure 1), particularly ceramides, hexosylceramides, and some phospholipids, based on a comparison of CHIKV-infected and VSV-infected samples. It should be mentioned that many of the fold changes supporting this choice (Figure 1C) remain below 1.5-fold, especially for ceramides, raising concerns about their biological significance, especially in absence of statistical analysis.While this strategy to compare CHIKV-infected and VSV-infected cells may allow for the identification of lipid change unique to CHIKV, as the authors mention, lipid alterations common to CHIKV and VSV could also play a crucial role in the infection process. In its current form, the manuscript does not enable assessing these shared variations. The authors should therefore present a more comprehensive analysis of all lipid classes detected by HILIC-IM-IS, accompanied by a robust statistical validation, similar to their previously published work on VSV-infected cells (Havranek et al., Viruses 2022, 14(1), 3).
Regarding the choice of Vero cells as a model, while African green monkey cell line is commonly used for viral propagation, its inability to mount an interferon response due to a genomic deletion raises important concerns. This deficiency is likely to impact both the lipidome imprinting observed upon CHIKV infection and the inhibitor-based functional analyses. Indeed, by modifying the dynamics of viral replication, the absence of interferon may lead to an atypical accumulation or alteration of lipids species. Moreover, interferon secretion and associated signaling are known to directly influence the synthesis and composition of many lipid classes, particularly phospholipids and sphingolipids. Ceramides, for instance, is significantly affected through the deregulation of ceramide metabolism-associated enzymes. This raises the question of how the observed results can be extrapolated to interferon-competent cells, especially human cells. Addressing this limitation by validating some key results in interferon-competent human cells, would enhance the study’s relevance.
Author Response
We added additional data showing UGT8-In19 also decreased CHIKV replication in hepatocytes an reduced titers. Now shown in figure 10. We also added relative abundances to figure 1.